# Adaptive Cross-Modal Few-Shot Learning

**Chen Xing**
Nankai University, Tianjin, China
Element AI, Montreal, Canada
`xingchen1113@gmail.com`

**Negar Rostamzadeh, Boris N. Oreshkin & Pedro O. Pinheiro**
Element AI, Montreal, Canada

## Abstract

Metric-based meta-learning techniques have successfully been applied to few-shot classification problems. However, leveraging cross-modal information in a few-shot setting has yet to be explored. When the support from visual information is limited in few-shot image classification, semantic representations (learned from unsupervised text corpora) can provide strong prior knowledge and context to help learning. Based on this intuition, we design a model that is able to leverage visual and semantic features in the context of few-shot classification. We propose an adaptive mechanism that is able to effectively combine both modalities conditioned on categories. Through a series of experiments, we show that our method boosts the performance of metric-based approaches by effectively exploiting language structure. Using this extra modality, our model bypass current unimodal state-of-the-art methods by a large margin on *mini*ImageNet and *tiered*ImageNet. The improvement in performance is particularly large when the number of shots are small.

## 1 Introduction

Deep learning methods have achieved major advances in areas such as speech, language and vision (LeCun et al., 2015). These systems, however, usually require a large amount of labeled data, which can be impractical or expensive to acquire. On the other hand, existing evidence suggests that the human visual system is capable of effectively operating in small data regime: humans can learn new concepts from a very few samples, by leveraging prior knowledge and context (Landau et al., 1988; Markman, 1991; Smith & Slone, 2017). The problem of learning new concepts with small number of labeled data points is usually referred to as *few-shot learning* (Bart & Ullman, 2005; Fink, 2005; Li et al., 2006; Lake et al., 2011). Most approaches addressing this problem are based on the *meta-learning* paradigm (Schmidhuber, 1987; Bengio et al., 1992; Thrun, 1998; Hochreiter et al., 2001), a class of algorithms and models focusing on learning how to (quickly) learn new concepts.

Recent progress in few-shot classification has primarily been made in the context of unimodal learning. In contrast to this, strong evidence supports the hypothesis that language helps the learning of new concepts in toddlers (Jackendoff, 1987; Smith & Gasser, 2005). This suggests that semantic features can be a powerful source of information in the context of few-shot image classification. While there have been many applications in combining visual and semantic embeddings to improve visual recognition tasks (*e.g.* in zero-shot learning (Frome et al., 2013) or image retrieval (Weston et al., 2011)), exploiting semantic language structure in the meta-learning framework has been mostly unexplored.

In this paper, we argue that few-shot classification can be considerably improved by leveraging semantic information from labels (learned, for example, from unsupervised text corpora). Visual and semantic feature spaces have heterogeneous structures by definition. For certain concepts, visual features might be richer and more discriminative than semantic ones. While for others, the inverse might be true. Moreover, when the support from visual information is limited, semantic features can provide strong prior knowledge and context to help learning. Based on this idea, we propose *Adaptive Modality Mixture Mechanism* (AM3), an approach that effectively and adaptively combines information from visual and semantic spaces.

AM3 is built on top of metric-based meta-learning approaches . These approaches perform classification by comparing distances in a learned metric space (from visual data). Different from previous metric-based methods, our model is able to exploit both visual and semantic feature spaces for classification. The semantic representation is learned from unsupervised text corpora and is easy to acquire. Our proposed mechanism performs classification in a feature space that is a convex combination of the two modalities. Moreover, we design the mixing coefficient to be adaptive w.r.t. different categories. Empirically, we show that our approach achieves considerable boost in performance over different metric-based meta-learning approaches on different number of shots. Moreover, our method is able to beat by a large margin current (single-modality) state of the art.

## 2 RELATED WORK

Many deep meta-learning approaches have been proposed to address few-shot learning problem. These methods can be roughly divided into two main types: metric-based and gradient-based approaches. Metric-based approaches aim at learning representations that minimize intra-class distances while maximizing the distance between different classes. These approaches tend to rely on an episodic training framework: the model is trained with sub-tasks (episodes) in which there are only a few training samples for each category. For example, in prototypical networks (Snell et al., 2017), a metric space is learned where embeddings of queries of one category are close to the centroid (or prototype) of supports of the same category, and far away from centroids of other classes in the episode. Due to simplicity and performance of this approach, many methods extended this work (Ren et al., 2018; Wang et al., 2018; Oreshkin et al., 2018; Dumoulin et al., 2018). Gradient-based meta-learning methods aim at training models that can generalize well to new tasks with only a few fine-tuning updates. Most these methods are built on top of model-agnostic meta-learning (MAML) framework (Finn et al., 2017). Given the universality of MAML, many follow-up works were recently proposed to improve its performance on few-shot learning (Nichol et al., 2018; Lacoste et al., 2017; Jiang et al., 2019; Rusu et al., 2019).

Current approaches mentioned above rely solely on visual features for few-shot classification. Our contribution is orthogonal to current metric-based approaches and can be integrated into them to boost performance in few-shot classification.

## 3 METHOD

### 3.1 PRELIMINARIES

**Episodic Training**   In few-shot learning, the class sets are disjoint between $\mathcal{D}_{\text{train}}$ and $\mathcal{D}_{\text{test}}$. The test set has only a few labeled samples per category. Most successful approaches rely on an *episodic* training paradigm: the few shot regime faced at test time is simulated by sampling small samples from the large labeled set $\mathcal{D}_{\text{train}}$ during training.

In general, models are trained on $K$-shot, $N$-way episodes. Each episode $e$ is created by first sampling $N$ categories from the training set and then sampling two sets of images from these categories: (i) the *support* set $\mathcal{S}_e = \{(s_i, y_i)\}_{i=1}^{N \times K}$ containing $K$ examples for each of the $N$ categories and (ii) the *query* set $\mathcal{Q}_e = \{(q_j, y_j)\}_{j=1}^{Q}$ containing different examples from the same $N$ categories.

The episodic training for few-shot classification is achieved by minimizing, for each episode, the loss of the prediction on samples in query set, given the support set:

$$\mathcal{L}(\theta) = \mathop{\mathbb{E}}_{(\mathcal{S}_e, \mathcal{Q}_e)} -\sum_{t=1}^{Q} \log p_\theta(y_t | q_t, \mathcal{S}_e) \,, \tag{1}$$

where $(q_t, y_t) \in \mathcal{Q}_e$ and $\mathcal{S}_e$ are, respectively, the sampled query and support set at episode $e$ and $\theta$ are the parameters of the model.

**Prototypical Networks**   We build our model on top of metric-based meta-learning methods. We chose prototypical network (Snell et al., 2017) for explaining our model due to its simplicity. We note, however, that the proposed method can potentially be applied to any metric-based approach.

Prototypical networks use the support set to compute a centroid (prototype) for each category (in the sampled episode) and query samples are classified based on the distance to each prototype. The model is a convolutional neural network (Lecun et al., 1998) $f : \mathbb{R}^{n_v} \rightarrow \mathbb{R}^{n_p}$, parameterized by $\theta_f$, that learns a $n_p$-dimensional space where samples of the same category are close and those of different categories are far apart.

For every episode $e$, each embedding prototype $p_c$ (of category $c$) is computed by averaging the embeddings of all support samples of class $c$:

$$\mathbf{p}_c = \frac{1}{|S_e^c|} \sum_{(s_i, y_i) \in \mathcal{S}_e^c} f(s_i) , \qquad (2)$$

where $\mathcal{S}_e^c \subset \mathcal{S}_e$ is the subset of support belonging to class $c$.

The model produces a distribution over the $N$ categories of the episode based on a softmax (Bridle, 1990) over (negative) distances $d$ of the embedding of the query $q_t$ (from category $c$) to the embedded prototypes:

$$p(y = c|q_t, S_e, \theta) = \frac{\exp(-d(f(q_t), \mathbf{p}_c))}{\sum_k \exp(-d(f(q_t), \mathbf{p}_k))} . \qquad (3)$$

We consider $d$ to be the Euclidean distance. The model is trained by minimizing Equation 1 and the parameters are updated with stochastic gradient descent.

## 3.2 ADAPTIVE MODALITY MIXTURE MECHANISM

The information contained in semantic concepts can significantly differ from visual information content. For instance, 'Siberian husky' and 'wolf', or 'komondor' and 'mop', might be difficult to discriminate with visual features, but might be easier to discriminate with language semantic features.

In the few-shot learning scenario, we hypothesize that both visual and semantic information can be useful for classification. Because we assume the visual and the semantic spaces have different structures, it is desirable that the proposed model exploit both modalities in the best way. We augment prototypical networks to incorporate language structure learned by a word-embedding model $\mathcal{W}$ (pre-trained on unsupervised large text corpora), containing label embeddings of all categories in $\mathcal{D}_{\text{train}} \cup \mathcal{D}_{\text{test}}$. In our model, we modify the prototype representation of each category by taking into account their label embeddings.

More specifically, we model the new prototype representation as a convex combination of the two modalities. That is, for each category $c$, the new prototype is computed as:

$$\mathbf{p}'_c = \lambda_c \cdot \mathbf{p}_c + (1 - \lambda_c) \cdot \mathbf{w}_c , \qquad (4)$$

where $\lambda_c$ is the *adaptive mixture coefficient* (conditioned on the category) and $\mathbf{w}_c = g(\mathbf{e}_c)$ is a transformed version of the label embedding for class $c$. This transformation $g : \mathbb{R}^{n_w} \rightarrow \mathbb{R}^{n_p}$, parameterized by $\theta_g$, is important to guarantee that both modalities lie on the space $\mathbb{R}^{n_p}$ of the same dimension and can be combined.

There are many different ways to adaptively calculate $\lambda_c$ to mix the two modalities. In this work we chose to condition the mixing coefficient on different categories. A very structured semantic space is a good choice for conditioning. Therefore, we chose a simple model for modulation conditioned on the semantic embedding space:

$$\lambda_c = \frac{1}{1 + \exp(-h(\mathbf{w}_c))} , \qquad (5)$$

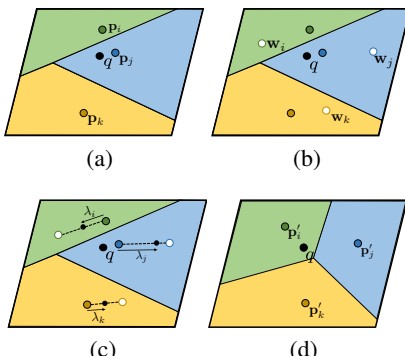

(a)    (b)

(c)    (d)

Figure 1: Qualitative example of how AM3 works. Assume query sample $q$ has category $i$. (a) The closest visual prototype to the query sample $q$ is $\mathbf{p}_j$. (b) The semantic prototypes. (c) The mixture mechanism modify the positions of the prototypes, given the semantic embeddings. (d) After the update, the closest prototype to the query is now the one of the category $i$, correcting the classification.

where $h$ is the adaptive mixing network, with parameters $\theta_h$. The mixing coefficient can be conditioned on different variables.

Table 1: Few-shot classification accuracy on *test* split of *mini*ImageNet and *tiered*ImageNet. Results in the top use only visual features. Cross-modal baselines are shown on the middle and our results (and their backbones) on the bottom part.

| Model | Test Accuracy | | | |
|---|---|---|---|---|
| | *mini*ImageNet | | *tiered*ImageNet | |
| | 1-shot | 5-shot | 1-shot | 5-shot |
| MatchingNets (Vinyals et al., 2016) | $43.56 \pm 0.84\%$ | $55.31 \pm 0.73\%$ | - | - |
| PrototypicalNets (Snell et al., 2017) | $49.42 \pm 0.78\%$ | $68.20 \pm 0.66\%$ | - | - |
| Disc-k-shot (Bauer et al., 2017) | $56.30 \pm 0.40\%$ | $73.90 \pm 0.30\%$ | - | - |
| (Ravi & Larochelle, 2017) | $43.44 \pm 0.77\%$ | $60.60 \pm 0.71\%$ | - | - |
| Meta-SGD (Li et al., 2017) | $50.47 \pm 1.87\%$ | $64.03 \pm 0.94\%$ | - | - |
| MAML (Finn et al., 2017) | $48.70 \pm 1.84\%$ | $63.11 \pm 0.92\%$ | $51.67 \pm 1.81\%$ | $70.30 \pm 0.08\%$ |
| Proto-k-Means (Ren et al., 2018) | $50.41 \pm 0.31\%$ | $69.88 \pm 0.20\%$ | $53.31 \pm 0.89\%$ | $72.69 \pm 0.74\%$ |
| SNAIL (Mishra et al., 2018) | $55.71 \pm 0.99\%$ | $68.80 \pm 0.92\%$ | - | - |
| CAML (Jiang et al., 2019) | $59.23 \pm 0.99\%$ | $72.35 \pm 0.71\%$ | - | - |
| LEO (Rusu et al., 2019) | $61.76 \pm 0.08\%$ | $77.59 \pm 0.12\%$ | $66.33 \pm 0.05\%$ | $81.44 \pm 0.09\%$ |
| ProtoNets++ | $56.52 \pm 0.45\%$ | $74.28 \pm 0.20\%$ | $58.47 \pm 0.64\%$ | $78.41 \pm 0.41\%$ |
| AM3-ProtoNets++ | $65.21 \pm 0.30\%$ | $75.20 \pm 0.27\%$ | $67.23 \pm 0.34\%$ | $78.95 \pm 0.22\%$ |
| TADAM (Oreshkin et al., 2018) | $58.56 \pm 0.39\%$ | $76.65 \pm 0.38\%$ | $62.13 \pm 0.31\%$ | $81.92 \pm 0.30\%$ |
| AM3-TADAM | $\mathbf{65.30 \pm 0.49\%}$ | $\mathbf{78.10 \pm 0.36\%}$ | $\mathbf{69.08 \pm 0.47\%}$ | $\mathbf{82.58 \pm 0.31\%}$ |

The training procedure is similar to that of the original prototypical networks. However, the distances $d$ (used to calculate the distribution over classes for every image query) are between the query and the cross-modal prototype $\mathbf{p}'_c$ Once again, the model is trained by minimizing Equation 1. Note that in this case the probability is also conditioned on the word embeddings $\mathcal{W}$. Figure 1 illustrates an example on how the proposed method works.

## 4 EXPERIMENTS

### 4.1 DATASETS

*mini*ImageNet is a subset of ImageNet ILSVRC12 dataset (Russakovsky et al., 2015). It contains 100 randomly sampled categories, each with 600 images. For fair comparison with other methods, we use the same split proposed by Ravi & Larochelle (2017), which contains 64 categories for training, 16 for validation and 20 for test. *tiered*ImageNet (Ren et al., 2018) is a larger subset of ImageNet.

We use GloVe (Pennington et al., 2014) to extract the semantic embeddings for the category labels. GloVe is an unsupervised approach based on word-word co-occurrence statistics from large text corpora. We use the Common Crawl version trained on 840B tokens. We also experimented with fastText embeddings (Joulin et al., 2016) and observed similar performances.

### 4.2 COMPARISON TO OTHER METHODS

Table 1 shows classification accuracy on *mini*ImageNet and *tiered*ImageNet. In the top part of each table, we show recent methods exploiting only visual features and at the bottom we show results of our method, AM3, with two different backbone architectures: ProtoNets++ and TADAM. We conclude multiple results from these experiments. First, our approach outperforms its backbone methods by a large margin in all cases tested. This indicates that language can be effectively leveraged to boost performance in classification with low number of shots. Second, AM3 (with TADAM backbone) achieves results superior to current (single modality) state of the art (Rusu et al., 2019). The margin in performance is particularly remarkable in the 1-shot scenario. Although our approach exploits semantic embeddings, we note they were learned with unlabeled text corpora. Third, it is clear that the gap between AM3 and the corresponding backbone gets reduced as the number of shots increases.

## 5 CONCLUSION

In this paper, we propose a method that can efficiently and effectively leverage cross-modal information for few-shot classification. Our method, AM3, adaptively combines visual and semantic features, given instance categories. AM3 boosts the performance of metric-based approaches by a large margin on different datasets and settings. Moreover, by leveraging unsupervised textual data, AM3 outperforms current unimodal state of the art on few-shot classification by a large margin. We also show that the semantic features are particularly helpful on the very low (visual) data regime (*e.g.* one-shot).

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
