# OpenReview forum: "Adaptive Cross-Modal Few-Shot Learning"
_ICLR.cc/2019/Workshop/LLD — LLD 2019_

### Official Review · AnonReviewer1 · 2019-04-07
**Interesting new setup and model, but lacking important model details**

**Rating:** 3
**Confidence:** 3

**Review:**

Summary
This paper proposes an approach for leveraging additional semantic information for solving the recently well-studied task of (visual) few-shot classification.

The authors demonstrate their particular approach based on the Prototypical Network model. This model is meta-learned via a sequence of tasks, with the goal in each being to correctly classify a set of ‘query’ images belonging to one of N classes after conditioning on a small handful of ‘support’ images from the same classes. Prototypical Networks’ mechanism of conditioning on the support set is to use it in order to create a prototype per class by averaging the corresponding support examples. Each query is then classified as the label of its nearest prototype. In this work, the prototype computation is modified to include an additional source of information: the word embedding of the corresponding class label. These two sources are combined via a convex combination with a learnable coefficient to decide the strength of each source. For the prototype of a particular class, this coefficient is defined as a sigmoid of the (transformed in a learnable way) word embedding of that class’ label.

They show experimentally that a particular variant of their proposed approach is able to surpass the (single modality) state-of-the-art on mini-ImageNet and tiered-ImageNet, with the greatest gains obtained in the 1-shot case.

Review
Pros:
[+] The proposed problem is an appealing one to study, since in the non-low-shot scenario, analogous multi-modal approaches have shown to be beneficial. Further, the fact that the semantic information is obtained in an unsupervised fashion from word co-occurrences in text corpora makes this setup attractive as no additional labels are required.
[+] The positive experimental results indeed confirm that semantic embeddings offer useful complementary information and can aid in visual few-shot meta-learning.

Cons:
[-] While I understand that 4 pages is very little space, I found some important information missing pertaining to the models that are proposed and being compared here. In particular, I wasn’t sure what ProtoNets++ is (no citation or explanation is included). Further, it seems that they implemented this approach on top of both ProtoNets++ (yielding AM3-ProtoNets++) and TADAM (yielding AM3-TADAM). I assume that the model they describe is the former. While I am familiar with TADAM, it’s not obvious to me how exactly the semantic information is incorporated into that model. I feel it’s better to sacrifice some space on a short explanation of this and cut space from somewhere else instead.
[-] Another concern is regarding the potential leakage of information from the meta-test set into the meta-training phase through the word embeddings. Specifically, during the training of word embeddings on large corpora, it may be that the statistics of occurrence of words whose labels belong to the meta-test set of the visual task had influenced the shaping of the word embeddings of words whose labels are in the meta-training set. I understand that this might be hard to control, and I’m not sure how large of a leakage effect there would be, but it would be useful to comment on it.

Overall, I feel this is an interesting problem, and this seems to be an interesting approach for addressing it, so I will recommend acceptance. In future experiments it would be interesting to address situations where not all visual concepts have associated word embeddings. I’m also curious if somehow episodically fine-tuning these word embeddings could yield additional performance gains.

---

### Official Review · AnonReviewer2 · 2019-04-08
**Interesting idea, but not the right experiments to prove the point**

**Rating:** 2
**Confidence:** 2

**Review:**

This paper proposes an approach for performing multi-modal few shot learning. The main contribution is a new way of combining visual (images) and text features (word embeddings derived from text) which enables the use of existing meta-learning approaches. According to the authors, there are no other multi-modal approaches for few-shot classification.

Pros:
- the problem seems important and useful.
- the entire paper is described clearly.

Cons:
- this paper is not really doing few-shot learning, because according to section 3.2. and the experiments, the authors use the test labels in order to know which word embeddings to assign to each sample: "[...] containing label embeddings of all categories in D_train ∪ D_test". In other words, the authors use the labels (which are the goal of the classification task) to find the match between the two input modalities (to know what Glove vector to assign to each image).
- the experiments compare the results only between this multimodal approach and visual approaches. I believe using the Glove embeddings alone (no visual input) could give very good results on their own, and it is thus crucial for the authors to compare with this scenario too.
- the explanation for why you chose this form for lambda_c is unclear: "A very structured semantic space is a good choice for conditioning."

Overall conclusion:
While the tackled problem is important and the paper is very well written, I believe the setting and the chosen dataset are artificial, because it requires looking at the test labels to create their inputs. If the authors had chosen a different dataset where text and images come naturally together (e.g., image captioning tasks), this would indeed be a good contribution. Moreover, I believe the experiments do not cover an important setting that the authors should have compared with (i.e. using only word embeddings as input), to prove that their method gains benefits from both modalities.

---

### Decision · Program_Chairs · 2019-04-16
**Acceptance Decision**

Accept